# Rethinking the Variational Interpretation of Accelerated Optimization Methods

**Peiyuan Zhang**[*]
ETH Zurich
talantyeri@gmail.com

**Antonio Orvieto**[*]
ETH Zurich
aorvieto@ethz.ch

**Hadi Daneshmand**
Inria Paris
seyed.daneshmand@inria.fr

## Abstract

The continuous-time model of Nesterov's momentum provides a thought-provoking perspective for understanding the nature of the acceleration phenomenon in convex optimization. One of the main ideas in this line of research comes from the field of classical mechanics and proposes to link Nesterov's trajectory to the solution of a set of Euler-Lagrange equations relative to the so-called Bregman Lagrangian. In the last years, this approach led to the discovery of many new (stochastic) accelerated algorithms and provided a solid theoretical foundation for the design of structure-preserving accelerated methods. In this work, we revisit this idea and provide an in-depth analysis of the action relative to the Bregman Lagrangian from the point of view of calculus of variations. Our main finding is that, while Nesterov's method is a stationary point for the action, it is often not a minimizer but instead a saddle point for this functional in the space of differentiable curves. This finding challenges the main intuition behind the variational interpretation of Nesterov's method and provides additional insights into the intriguing geometry of accelerated paths.

## 1 Introduction

This paper focuses on the problem of unconstrained convex optimization, i.e. to find

$$x^* \in \arg\min_{x \in \mathbb{R}^d} f(x), \tag{P}$$

for some lower bounded convex $L$-smooth[2] loss $f \in \mathcal{C}^1(\mathbb{R}^d, \mathbb{R})$.

**Nesterov's acceleration.** Nemirovskii and Yudin (1983) showed that no gradient-based optimizer can converge to a solution of (P) faster than $\mathcal{O}(k^{-2})$, where $k$ is the number of gradient evaluations[3]. While Gradient Descent (GD) converges like $\mathcal{O}(k^{-1})$, the optimal rate $\mathcal{O}(k^{-2})$ is achieved by the celebrated Accelerated Gradient Descent (AGD) method, proposed by Nesterov (1983):

$$x_{k+1} = y_k - \eta\nabla f(y_k), \quad \text{with} \quad y_k = x_k + \frac{k-1}{k+2}(x_k - x_{k-1}). \tag{AGD}$$

The intuition behind Nesterov's method and the fundamental reason behind acceleration is, to this day, an active area of research (Allen-Zhu and Orecchia, 2014; Defazio, 2019; Ahn, 2020).

**ODE models.** Towards understanding the acceleration mechanism, Su et al. (2016) made an interesting observation: the convergence rate gap between GD and AGD is retained in the continuous-time limits (as the step-size $\eta$ vanishes):

$$\dot{X} + \nabla f(X) = 0 \quad \text{(GD-ODE)}, \qquad \ddot{X} + \frac{3}{t}\dot{X} + \nabla f(X) = 0 \quad \text{(AGD-ODE)}$$

---

[*]Equal Contribution.

[2]A differentiable function $f : \mathbb{R}^d \to \mathbb{R}$ is said to be $\beta$-smooth if it has $\beta$-Lipschitz gradients.

[3]This lower bound holds just for $k < d$ hence it is only interesting in the high-dimensional setting.

35th Conference on Neural Information Processing Systems (NeurIPS 2021).

where $\dot{X} := dX/dt$ denotes the time derivative (velocity) and $\ddot{X} := d^2X/dt^2$ the acceleration. Namely, we have that GD-ODE converges like $\mathcal{O}(t^{-1})$ and AGD-ODE like $\mathcal{O}(t^{-2})$, where $t > 0$ is the time variable. This seminal paper gave researchers a new tool to understand the nature of accelerated optimizers through Bessel Functions (Su et al., 2016), and led to the design of many novel fast and interpretable algorithms outside the Euclidean setting (Wibisono et al., 2016; Wilson et al., 2019), in the stochastic setting (Krichene et al., 2015; Xu et al., 2018) and also in the manifold setting (Alimisis et al., 2020; Duruisseaux and Leok, 2021).

**Nesterov as solution to Euler-Lagrange equations.** It is easy to see that AGD-ODE can be recovered from Euler-Lagrange equations, starting from the time-dependent *Lagrangian*

$$L(X, \dot{X}, t) = t^3 \left( \frac{1}{2} \|\dot{X}\|^2 - f(X) \right). \tag{1}$$

Indeed, the Euler-Lagrange equation

$$\frac{d}{dt} \left( \frac{\partial}{\partial \dot{X}} L(X, \dot{X}, t) \right) = \frac{\partial}{\partial X} L(X, \dot{X}, t) \tag{2}$$

reduces in this case to $t^3 \ddot{X} + 3t^2 \dot{X} + t^3 \nabla f(X) = 0$, which is equivalent to AGD-ODE (assuming $t > 0$). In a recent influential paper, Wibisono et al. (2016) generalized the derivation above to non-Euclidean spaces, where the degree of separation between points $x$ and $y$ is measured by means of the *Bregman Divergence* (Bregman, 1967) $D_\psi(x, y) = \psi(y) - \psi(x) - \langle \nabla \psi(x), y - x \rangle$, where $\psi : \mathbb{R}^d \to \mathbb{R}$ is a strictly convex and continuously differentiable function (see e.g. Chapter 1.3.2 in Amari (2016)). Namely, they introduced the so-called *Bregman Lagrangian*:

$$L_{\alpha,\beta,\gamma}(X, \dot{X}, t) = e^{\alpha(t)+\gamma(t)} \left( D_\psi(X + e^{-\alpha(t)}V, X) - e^{\beta(t)}f(X) \right), \tag{3}$$

where $\alpha, \beta, \gamma$ are continuously differentiable functions of time. The Euler-Lagrange equations imply

$$\ddot{X} + (e^{\alpha(t)} - \dot{\alpha}(t))\dot{X} + e^{2\alpha(t)+\beta(t)} \left[ \nabla^2 \psi(X + e^{-\alpha(t)}\dot{X}) \right]^{-1} \nabla f(X) = 0. \tag{4}$$

The main result of Wibisono et al. (2016) is that, under the *ideal-scaling conditions* $\dot{\beta}(t) \leq e^{\alpha(t)}$ and $\dot{\gamma}(t) = e^{\alpha(t)}$, any solution to Eq. (4) converges to a solution of (P) at the rate $\mathcal{O}(e^{-\beta(t)})$. Under the choice $\psi(x) = \frac{1}{2}\|x\|_2^2$, we get back to the Euclidean metric $D_\psi(x, y) = \frac{1}{2}\|x - y\|_2^2$. Moreover, choosing $\alpha(t) = \log(2/t)$, $\beta(t) = \gamma(t) = 2\log(t)$, we recover the original Lagrangian in Eq. (1) and $\mathcal{O}(e^{-\beta(t)}) = \mathcal{O}(t^{-2})$, as derived in Su et al. (2016).

**Impact of the variational formulation.** The variational formulation in Wibisono et al. (2016) has had a considerable impact on the recent developments in the theory of accelerated methods. Indeed, this approach can be used to design and analyze new accelerated algorithms. For instance, Xu et al. (2018) used the Lagrangian mechanics formalism to derive a novel simplified variant of accelerated stochastic mirror descent. Similarly, França et al. (2021), Muehlebach and Jordan (2021) used the dual Hamiltonian formalism to study the link between symplectic integration of dissipative ODEs and acceleration. Due to its rising importance in the field of optimization, the topic was also presented by Prof. M. I. Jordan as a plenary lecture at the *International Congress of Mathematicians* in 2018 (Jordan, 2018), centered around the question "*what is the optimal way to optimize?*".

**Imprecise implications of the variational formulation.** While the Lagrangian formalism has been inspiring and successful for algorithm design and analysis, its precise implications for the geometry and the path of accelerated solutions have not been examined in a mathematically rigorous way (to the best of our knowledge). In Jordan (2018) it is hinted that, since Nesterov's method solves the Euler-Lagrange equations, it **minimizes the action** functional $\int_{t_1}^{t_2} L_{\alpha,\beta,\gamma}(Y, \dot{Y}, t)dt$ over the space of curves by the *minimum action principle* of classical mechanics (Arnol'd, 2013). This claim[4] is inaccurate. Indeed, the term *minimum action principle* is misleading[5]: solving Euler-Lagrange

---

[4]Paragraph before Eq. (9) in Jordan (2018): "*[...] we use standard calculus of variations to obtain a differential equation whose solution **is the path that optimizes** the time-integrated Bregman Lagrangian*".

[5]From Section 36.2 in Gelfand and Fomin (2000): "*The principle of least action is widely used [...]. However, in a certain sense the principle is not quite true [...]. We shall henceforth replace the principle of least action by the principle of stationary action. In other words, the actual trajectory of a given mechanical system **will not be required to minimize the action** but only to cause its first variation to vanish.*"

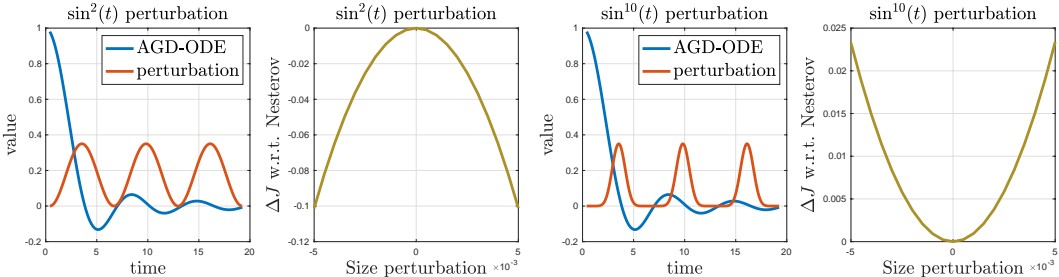

Figure 1: Optimization of $f(x) = x^2/2$ using AGD-ODE. Peturbations (vanishing at extrema) are added to the AGD-ODE solution: depending on the perturbation kind (i.e. direction in space of curves), the local behavior is either a max or a min. Hence, Nesterov's path can be a saddle point for the action (formally shown in Sec. 3.1).

only makes the action stationary (necessary condition: vanishing first-order derivative), but does not guarantee minimality — this only holds in physics for very special cases[6], which do not include even simple mechanical systems like the pendulum (proof in Section 36.2 of Gelfand and Fomin (2000)). Indeed, from a theoretical perspective, the claim requires computing the second variation along Nesterov's path. Quite surprisingly, even though many papers are dedicated to the variational formulation (Wibisono et al., 2016; Jordan, 2018; Casgrain, 2019; Duruisseaux and Leok, 2021), to the best of our knowledge there is no work which provides an in-depth rigorous study of the action relative to Bregman Lagrangian and that characterizes minimality of Nesterov in the space of curves.

**Our contributions.** Intrigued by the non-trivial open question of minimality of Nesterov's path and by the enigmatic geometry of accelerated flows, in this paper, we examine the properties of accelerated gradient methods from the perspective of calculus of variations.

1. In Sec. 3 we study the minimality of classical Nesterov's ODE (damping $3/t$) proposed by Su et al. (2016) on multidimensional quadratic losses. By using Jacobi's theory for the second variation (summarized in Sec. 2), we find that Nesterov's path is optimal only if the integration interval $[t_1, t_2]$ is small enough. In contrast, if $t_2 - t_1 > \sqrt{40/\beta}$ ($\beta$ is Lipschitz constant for the gradient), Nesterov's path is actually a **saddle point** for the action (see Fig. 1).

2. In Sec. 4 we extend the analysis to the $\mu$-strongly convex setting and thus consider a constant damping $\alpha$. We show that, for extremely overdamped Nesterov flows ($\alpha \geq 2\sqrt{\beta}$), i.e for highly suboptimal parameter tuning (acceleration holds only for $\alpha \approx 2\sqrt{\mu}$), Nesterov's path is always a minimizer for the action. In contrast, we show that for $\alpha < 2\sqrt{\beta}$ (acceleration setting), if $t_2 - t_1 > 2\pi/\sqrt{4\beta - \alpha}$, Nesterov's path is again a saddle point for the action.

3. In Sec. 5 we discuss the implications of our results for the theory of accelerated methods and propose a few interesting directions for future research.

We start by recalling some definitions and results from calculus of variations, which we adapt from classical textbooks (Landau and Lifshitz, 1976; Arnol'd, 2013; Gelfand and Fomin, 2000).

## 2   Background on calculus of variations

We work on the vector space of curves $\mathcal{C}^1([t_1, t_2], \mathbb{R}^d)$ with $t_1, t_2 \in [0, \infty)$. We equip this space with the standard norm $\|Y\| = \max_{t_1 \leq t \leq t_2} \|Y(t)\|_2 + \max_{t_1 \leq t \leq t_2} \|\dot{Y}(t)\|_2$. Under this choice, for any regular Lagrangian $L$, the action functional $J[Y] := \int_{t_1}^{t_2} L(Y, \dot{Y}, t)dt$ is continuous.

**First variation.** Let $D$ be the linear subspace of continuously differentiable displacements curves $h$ such that $h(t_1) = h(t_2) = 0$. The corresponding *increment* of $J$ at $Y$ along $h$ is defined as $\Delta J[Y; h] := J[Y + h] - J[Y]$. Suppose that we can write $\Delta J[Y; h] = \varphi[Y; h] + \epsilon\|h\|$, where $\varphi$ is linear in $h$ and $\epsilon \to 0$ as $\|h\| \to 0$. Then, $J$ is said to be differentiable at $Y$ and the linear functional $\delta J[Y; \cdot] : D \to \mathbb{R}$ such that $\delta J[Y; h] := \varphi[Y; h]$ is called *first variation* of $J$ at $Y$. It can be shown that, if $J$ is differentiable at $Y$, then its first variation at $Y$ is unique.

**Extrema.** $J$ is said to have an *extremum* at $Y$ if $\exists \delta > 0$ such that, $\forall h \in D$ with $\|h\| \leq \delta$, the sign of $J[Y + h] - J[Y]$ is constant. A *necessary* condition for $J$ to have an extremum at $Y$ is that

$$\delta J[Y; h] = 0, \quad \text{for all } h \in D. \tag{5}$$

---

[6]e.g. free particle in vanishing potentials, or $t_1 \approx t_2$, see Remark 2 at the end of Section 21 and Section 36.2.

We proceed with stating one of the most well-known results in calculus of variations, which follows by using Taylor's theorem on $J[Y] = \int_{t_1}^{t_2} L(Y, \dot{Y}, t)\, \mathrm{d}t$.

**Theorem 1** (Euler-Lagrange equation). *A necessary condition for the curve $Y \in \mathcal{C}^1([t_1, t_2], \mathbb{R}^d)$ to be an extremum for $J$ (w.r.t. $D$) is that it satisfies the Euler-Lagrange equations* (2).

It is crucial to note that Theorem 1 provides a *necessary, but not sufficient* condition for an extremum — indeed, the next paragraph is completely dedicated to this.

**Second Variation.** Thm. 1 does not distinguish between extrema (maxima or minima) and saddles. For this purpose, we need to look at the *second variation*.

Suppose that the increment of $J$ at $Y$ can be written as $\Delta J[Y; h] = \varphi_1[Y; h] + \varphi_2[Y; h] + \epsilon \|h\|^2$, where $\varphi_1$ is linear in $h$, $\varphi_2$ is quadratic in $h$ and $\epsilon \to 0$ as $\|h\| \to 0$. Then $J$ is said to be *twice differentiable* and the functional $\delta^2 J[Y :, \cdot] : D \to \mathbb{R}$ s.t. $\delta^2 J[Y, h] := \varphi_2[Y; h]$ is called the *second variation* of $J$ at $Y$. Uniqueness of second variation is proved in the same way as the first variation.

**Theorem 2.** *A necessary condition for the curve $Y \in \mathcal{C}^1([t_1, t_2], \mathbb{R}^d)$ to be a local minimum for $J$ (w.r.t $D$) is that it satisfies $\delta^2 J[Y; h] \geq 0$. For local maxima, the sign is flipped.*

**Jacobi equations.** Recall that $J[Y] = \int_{t_1}^{t_2} L(Y, \dot{Y}, t) dt$. Using the notation $L_{YZ} = \partial^2 L / (\partial Y \partial Z)$, the Taylor expansion for $\Delta J[Y; h] = J[Y + h] - J[Y]$ if $\|h\| \to 0$ converges to

$$\Delta J[Y; h] = \int_{t_1}^{t_2} \left( L_Y h + L_{\dot{Y}} \dot{h} \right) dt + \frac{1}{2} \int_{t_1}^{t_2} \left( L_{YY} h^2 + L_{\dot{Y}\dot{Y}} \dot{h}^2 + 2 L_{Y\dot{Y}} h \dot{h} \right) dt, \qquad (6)$$

where the equality holds coordinate-wise. Therefore, $\delta J[Y; h] = \int_{t_1}^{t_2} \left( L_Y h + L_{\dot{Y}} \dot{h} \right) dt$ and

$$\begin{aligned}
\delta^2 J[Y; h] &= \frac{1}{2} \int_{t_1}^{t_2} \left( L_{YY} h^2 + 2 L_{Y\dot{Y}} h \dot{h} + L_{\dot{Y}\dot{Y}} \dot{h}^2 \right) dt \\
&= \frac{1}{2} \int_{t_1}^{t_2} \left( L_{YY} - \frac{d}{dt} L_{Y\dot{Y}} \right) h^2 dt + \frac{1}{2} \int_{t_1}^{t_2} L_{\dot{Y}\dot{Y}} \dot{h}^2 dt \\
&= \frac{1}{2} \int_{t_1}^{t_2} \left( P \dot{h}^2 + Q h^2 \right) dt, \qquad (7)
\end{aligned}$$

where $P = L_{\dot{Y}\dot{Y}}$, $Q = L_{YY} - \frac{d}{dt} L_{Y\dot{Y}}$, and the second equality follows from integration by parts since $h$ vanishes at $t_1$ and $t_2$. Using this expression, it is possible to derive an easy necessary (but not sufficient) condition for minimality.

**Theorem 3** (Legendre's necessary condition). *A necessary condition for the curve $Y$ to be a minimum of $J$ is that $L_{\dot{Y}\dot{Y}}$ is positive semidefinite.*

**Conjugate points.** A crucial role in the behavior of $\delta^2 J[Y; h]$ is played by the shape of the solutions to Jacobi's differential equation $\frac{d}{dt}(P\dot{h}) - Qh = 0$. A point $t \in (t_1, t_2)$ is said to be *conjugate* to point $t_1$ (w.r.t. $J$) if Jacobi's equation admits a solution which vanishes at both $t_1$ and $t$ but is not identically zero. We have the following crucial result.

**Theorem 4** (Jacobi's condition). *Necessary and sufficient conditions for $Y$ to be a local minimum for $J$ are: (1) $Y$ satiesfies the Euler-Lagrange Equation; (2) $P$ positive definite; (3) $(t_1, t_2)$ contains no points conjugate to $t_1$.*

# 3 Analysis of the action of Nesterov's path with vanishing damping $3/t$

This section is dedicated to the analysis of the action functional relative to Eq. (3). By incorporating the tool of variational calculus, we study the optimality of Nesterov's method in minimizing the action. We start by a general abstract analysis in the convex quadratic case in Sec. 3.1, and then present an intuitive analytical computation in Sec. 3.2. The non-quadratic case is discussed in Sec. 3.4.

## 3.1 Solutions to Jacobi's equation for the Bregman Lagrangian in the quadratic setting

For the sake of clarity, we start by considering the Lagrangian in Eq. (1) for the simple one-dimensional case $f(x) = \beta x^2 / 2$. We have

$$Q = L_{YY} - \frac{d}{dt} L_{Y\dot{Y}} = -\beta t^3, \quad P = L_{\dot{Y}\dot{Y}} = t^3. \qquad (8)$$

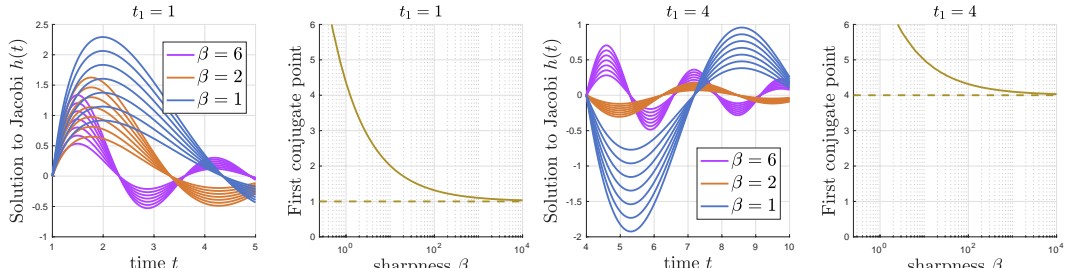

Figure 2: First conjugate point to $t_1 = 1, 4$ for quadratics $\beta x^2/2$, under the settings of Section 3.1. For each value of $\beta$, six solutions $h(t)$ to the Jacobi equation (each one has different velocity) are shown.

Therefore, Jacobi's equation relative to the action functional $\int_{t_1}^{t_2} L(Y, \dot{Y}, t) dt$ with $t_1 > 0$ is

$$\frac{d}{dt}(t^3 \dot{h}) - \beta t^3 h = 0. \implies t^3 \ddot{h} + 3t^2 \dot{h} + \beta t^3 h = 0 \implies \ddot{h} + \frac{3}{t} \dot{h} + \beta h = 0, \quad (9)$$

which is itself Nesterov's ODE. Following the procedure outlined in Theorem 2, we now study the solutions $h$ such that $h(t_1) = 0$. Any solution to Eq. (9) can be written as[7]

$$h(t) = C \frac{\mathcal{Y}_1(\sqrt{\beta}\, t)}{t} - C \frac{\mathcal{Y}_1(\sqrt{\beta}\, t_1)\, \mathcal{J}_1(\sqrt{\beta}\, t)}{\mathcal{J}_1(\sqrt{\beta}\, t_1)\, t}, \quad (10)$$

where $C > 0$ specifies the initial velocity (see Fig. 2), $\mathcal{J}_\alpha$ is the Bessel function of the first kind and $\mathcal{Y}_\alpha$ is the Bessel function of the second kind.

$$\mathcal{J}_\alpha(x) = \sum_{m=0}^{\infty} \frac{(-1)^m}{m!\,\Gamma(m+\alpha+1)} \left(\frac{x}{2}\right)^{2m+\alpha}, \quad \mathcal{Y}_\alpha(x) = \frac{\mathcal{J}_\alpha(x)\cos(\alpha\pi) - \mathcal{J}_{-\alpha}(x)}{\sin(\alpha\pi)}. \quad (11)$$

Points $t > t_1$ conjugate to $t_1$ satisfy $h(t) = 0$, which results in the identity $\mathcal{Y}_1(\sqrt{\beta}\, t)/\mathcal{Y}_1(\sqrt{\beta}\, t_1) = \mathcal{J}_1(\sqrt{\beta}\, t)/\mathcal{J}_1(\sqrt{\beta}\, t_1)$. Remarkably, this condition does not depend on $C$, but only on $t_1$ and on the sharpness $\beta$. Let us now fix these parameters and name $K_{\beta,t_1} = \mathcal{Y}_1(\sqrt{\beta}\, t_1)/\mathcal{J}_1(\sqrt{\beta}\, t_1)$. Points conjugate to $t_1$ then satisfy $\mathcal{Y}_1(\sqrt{\beta}\, t) = K_{\beta,t_1} \mathcal{J}_1(\sqrt{\beta}\, t)$. We now recall the following expansions (Watson, 1995), also used by Su et al. (2016):

$$\mathcal{J}_1(x) = \sqrt{\frac{2}{\pi x}} \left( \cos\left( x - \frac{3\pi}{4} \right) + \mathcal{O}\left(\frac{1}{x}\right) \right), \quad \mathcal{Y}_1(x) = \sqrt{\frac{2}{\pi x}} \left( \sin\left( x - \frac{3\pi}{4} \right) + \mathcal{O}\left(\frac{1}{x}\right) \right). \quad (12)$$

Since $\mathcal{J}_1$ and $\mathcal{Y}_1$ asymptotically oscillate around zero and are out of synch ($\pi/2$ difference in phase), for $t$ big enough the condition $\mathcal{Y}_1(\sqrt{\beta}\, t) = K_{\beta,t_1} \mathcal{J}_1(\sqrt{\beta}\, t)$ is going to be satisfied. Further, this condition is going to be satisfied for a smaller value for $t$ if $\beta$ is increased, as confirmed by Figure 2.

---

**Theorem 5** (Local optimality of Nesterov with vanishing damping). *Let $f : \mathbb{R}^d \to \mathbb{R}$ be a convex quadratic, and let $X : \mathbb{R} \to \mathbb{R}^d$ be a solution to the ODE $\ddot{X} + \frac{3}{t}\dot{X} + \nabla f(X) = 0$. For $0 < t_1 < t_2$, consider the action functional $J[Y] = \int_{t_1}^{t_2} L(Y, \dot{Y}, t) dt$, mapping $Y \in \mathcal{C}^1([t_1, t_2], \mathbb{R}^d)$, to a real number. Then, if $|t_2 - t_1|$ is small enough, there are no points conjugate to $t_1$ and Nesterov's path minimizes $J$ over all curves such that $Y(t_1) = X(t_1)$ and $Y(t_2) = X(t_2)$. The length of the optimality interval $|t_2 - t_1|$ shrinks as $\beta$, the maximum eigenvalue of the Hessian of $f$, increases.*

*Proof.* The argument presented in this section can be lifted to the multidimensional case. Indeed, since the dynamics in phase space is linear, it's geometry is invariant to rotations and we can therefore assume the Hessian is diagonal. Next, Jacobi's equation has to be solved coordinate-wise, which leads to a logical AND between conjugacy conditions. By the arguments above, the dominating condition is the one relative to the maximum eigenvalue $\beta$. $\square$

The following corollary shows that Nesterov's path actually becomes suboptimal if the considered time interval is big enough. This is also verified numerically in Figure 1.

---

[7]Symbolic computations are checked in Maple/Mathematica, numerical simulations are performed in Matlab.

**Corollary 6** (Nesterov with vanishing damping is not globally optimal). *In the settings of Thm. 5, for $|t_2 - t_1|$ big enough, Nesterov's path becomes a saddle point for $J$.*

*Proof.* Non-existence of conjugate points is necessary and sufficient for minimality/maximality. □

In the next subsection, we provide a constructive proof for Cor. 6, which allows us to derive a concrete bound for $|t_2 - t_1|$. In particular, we show in Prop. 7 that, for the case of vanishing damping $3/t$, Nesterov's path is always a saddle point for the action if $|t_2 - t_1| > \sqrt{40/\beta}$.

**Optimality for the special case $X(t_2) = x^*$.** Some readers might have already realized that Jacobi's equation Eq. (9) in the quadratic potential case is itself the solution of Nesterov's ODE. This means that, if $t_1 \approx 0$, the first time conjugate to $t_1$ is exactly when *Nesterov's path reaches the minimizer*. Hence — only in the one-dimensional case — it is actually true that, if we do not consider arbitrary time intervals but

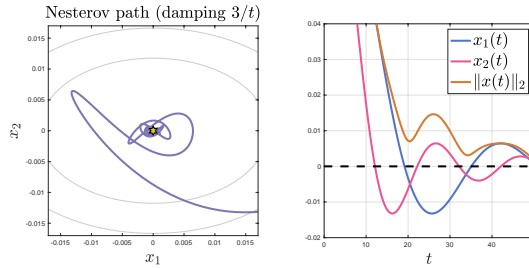

only the first interval before the solution first touches the optimizer, Nesterov's path always minimizes the action. Sadly, this strong and interesting result *is not valid in higher dimensions*, since Nesterov's path in general never actually crosses the minimizer in finite time [8].

**Remark on dropping boundary conditions.** The results in this section are formulated for the fixed boundaries case $Y(t_1) = X(t_1)$ and $Y(t_2) = X(t_2)$, where $X$ is the solution to Nesterov's ODE. From an optimization viewpoint, this requirement seems strong. Ideally, we would want $X(t_2)$ to be *any point close to the minimizer* (say inside an $\epsilon$-ball). A simple reasoning proves that Nesterov's path can be a saddle point for the action also in this case. By contradiction, assume Nesterov's trajectory minimizes the action among all curves that reach any point inside a small $\epsilon$-ball at time $t_2$. Then, Nesterov's path also minimizes the action in the (smaller) space of curves that reach exactly $X(t_2)$. By Cor. 6, this leads to a contradiction if $t_2$ is big enough.

## 3.2 A constructive proof of Nesterov's suboptimality in the quadratic setting

We now present a direct computation, to shed some light on the suboptimality of Nesterov's path in the context of Thm. 5. In the setting of Sec. 3.1, the second variation of $J$ along $\gamma$ is $\frac{1}{2}\int_{t_1}^{t_2} t^3 [\dot{h}(t)^2 - \beta h(t)^2]dt$, independent of $\gamma$. Consider now the finite-norm perturbation (vanishing at boundary):

$$\tilde{h}_{\epsilon,c}(t) = \begin{cases} 0 & t \leq c - \epsilon \text{ or } t \geq c + \epsilon \\ \frac{t-c+\epsilon}{\epsilon} & t \in (c-\epsilon, c) \\ \frac{c+\epsilon-t}{\epsilon} & t \in (c, c+\epsilon) \end{cases}, \quad c \in (t_1, t_2), \epsilon < \min(c - t_1, t_2 - c). \quad (13)$$

This is a triangular function with support $(c - \epsilon, c + \epsilon)$ and height one. Let $h_{\epsilon,c}$ be a $\mathcal{C}^1$ modification of $\tilde{h}_{\epsilon,c}$ such that $\|\tilde{h}_{\epsilon,c} - h_{\epsilon,c}\|$ is negligible [9]. For any scaling factor $\sigma > 0$,

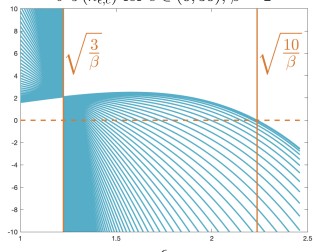

$$\delta^2 J(\sigma \cdot h_{\epsilon,c}) = -\sigma^2 \frac{\left(\frac{3\beta\epsilon^4}{10} + (\beta c^2 - 3)\epsilon^2 - 3c^2\right)c}{3\epsilon}. \quad (14)$$

The denominator is always positive. Hence, we just need to study the sign of the numerator, with respect to changes in $\epsilon > 0$ and $c > 0$. Consider for now $c$ fixed, then the zeros of the numerator are at $(15 - 5u \pm \sqrt{25u^2 - 60u + 225})/(3\beta)$, with $u := \beta c^2$. Since $25u^2 - 60u + 225 > 0$ for all $u \geq 0$, the solution has two real roots. However, only one root $\epsilon_*^2(u, \beta)$ is admissible, since the smallest one is always negative [10]. As a result, for fixed $c > 0$, $\delta^2 J(\sigma h_{\epsilon,c})$ changes sign only at $\epsilon_*^2(u, \beta)$. Note that $\epsilon_*^2(u, \beta)$ is decreasing as a function of $u$ and $\epsilon_*^2(0, \beta) = 10/\beta$ as well as $\lim_{u \to \infty} \epsilon_*^2(u, \beta) = 3/\beta$. Therefore, for any $c, \beta > 0$, we showed that $\delta^2 J(\sigma h_{\epsilon,c})$ changes sign when $\epsilon_* \in [\sqrt{3/\beta}, \sqrt{10/\beta}]$. If choosing

---

[8] The first crossing time in each direction depends on the sharpness in each direction, hence by the time each coordinate reaches zero, we already have a conjugate point.

[9] Standard technique in calculus of variation, see e.g. proof of Legendre's Thm (Gelfand and Fomin, 2000).

[10] If $u > 0$, then $15 - 5u - \sqrt{25u^2 - 60u + 225} < 0$.

$\epsilon$ big is allowed by the considered interval ($h_{\epsilon,c}$ has to vanish at $t_1, t_2$), then the second variation is indefinite. This happens if $|t_2 - t_1| > 2\epsilon_*$. By taking $\sigma \to 0$, we get the following result.

**Proposition 7** (Sufficient condition for saddle). *The second variation of the action of Nesterov's Lagrangian (damping $3/t$) on $f(x) = \beta x^2/2$ is an indefinite quadratic form for $|t_2 - t_1| > \sqrt{40/\beta}$. This result generalizes to $\beta$-smooth multidimensional convex quadratics.*

We remark that the inverse dependency on the square root of $\beta$ is also predicted by the general proof in Sec. 3.1, where the argument of the Bessel functions is always $t\sqrt{\beta}$.

### 3.3 Unboundedness of the action for large integration intervals (from above and below)

In Prop. 7, we showed that for big enough integration intervals, Nesterov's method with damping $3/t$ on $f(x) = \beta x^2/2$ is saddle point for the action. This suggests that the action is itself unbounded — both from above and below. It is easy to show this formally.

**Proposition 8** (Unboundedness of the action). *Let $L$ be Lagrangian of Nesterov's method with damping $3/t$ on a $\beta$-smooth convex quadratic and $J[Y] = \int_{t_1}^{t_2} L(Y, \dot{Y}, t)dt$. Let $a, b$ be two arbitrary vectors in $\mathbb{R}^d$. There exists a sequence of curves $(Y_k)_{k \in \mathbb{N}}$, with $Y_k \in \mathcal{C}^1([t_1, t_2], \mathbb{R}^d)$ and $Y_k(t_1) = a, Y_k(t_2) = b$ for all $k \in \mathbb{N}$, such that $J[Y_k] \xrightarrow{k} \infty$. In addition, if $|t_2 - t_1| > \sqrt{40/\beta}$ there exists another sequence with the same properties diverging to $-\infty$.*

*Proof.* The proof is based on the computation performed for Prop. 7. Crucially, note that for the quadratic loss function case we have $\delta^2 J = J$. For the case $a = b = 0$, we showed that for any interval $[t_1, t_2]$, by picking $\epsilon$ small enough, we have $J(h_{\epsilon,c}) = \delta^2 J(h_{\epsilon,c}) > 0$ (also illustrated in the figure supporting the proof). Hence, $J(\sigma \cdot h_{\epsilon,c}) \to +\infty$ as $\sigma \to \infty$. Same argument holds for $-\infty$ in the large interval case. This proves the assertion for $a = b = 0$. Note that the curves corresponding to the diverging sequences can be modified to start/end at any $a, b \in \mathbb{R}^d$ at the price of a bounded error in the action. This does not modify the behavior in the limit; hence, the result follows. □

### 3.4 Optimality of Nesterov with vanishing damping if curvature vanishes (polynomial loss)

Note that the bound on $|t_2 - t_1|$ in Prop. 7 gets loose as $\beta$ decreases. This is also predicted by the argument with Bessel functions in Sec. 3.1, and clear from the simulation in Fig. 2. As a result, as curvature vanishes, Nesterov's path becomes optimal for larger and larger time intervals. This setting is well described by polynomial losses $f(x) \propto (x - x^*)^p$, with $p > 2$. As Nesterov's path approaches the minimizer $x^*$, the curvature vanishes; hence, for every $\beta > 0$ there exists a time interval $(t_1, \infty)$ where the curvature along Nesterov's path is less then $\beta$. This suggest that, for losses with vanishing curvature at the solution, there exists a time interval $(t_*, \infty)$ where Nesterov's path is actually a minimizer for the action. While this claim is intuitive, it is extremely hard to prove formally since in this case the second variation of $J$ depends on the actual solution of Nesterov's equations — for which no closed-form formula is known in the polynomial case (Su et al., 2016).

However, we also note that the vanishing sharpness setting is only interesting from a theoretical perspective. Indeed, *regularized* machine learning objectives do not have this property. Actually, in the *deep neural network* setting, it is known that the sharpness (maximum eig. of the Hessian) actually increases overtime (Yao et al., 2020; Cohen et al., 2021). Hence, it is safe to claim that in the machine learning setting Nesterov's path is only optimal for small time intervals, as shown in Thm. 5.

## 4 Analysis of the action of Nesterov's path with constant damping $\alpha$

In the $\mu$-strongly convex case[11], it is well known that a constant damping $\alpha = 2\sqrt{\mu}$ yields acceleration compared to gradient descent[12]. This choice completely changes the geometry of Nesterov's path and needs a separate discussion. The corresponding Lagrangian is

$$L_\alpha(Y, \dot{Y}, t) = e^{\alpha t}\left(\frac{1}{2}\|\dot{Y}\|^2 - f(Y)\right). \tag{15}$$

---

[11]Hessian eigenvalues lower bounded by $\mu > 0$.

[12]The corresponding rate is linear and depends on $\sqrt{\mu/\beta}$, as opposed to $\mu/\beta$ (GD case).

Again, we consider the quadratic function $f(x) = \beta x^2/2$ and examine Jacobi's ODE $\ddot{h}(t) + \alpha\dot{h}(t) + \beta h(t) = 0$. We have to determine whether there exists a non-trivial solution such that $h(t_1) = h(t_2) = 0$ and $h(t)$ vanishes also at a point $t \in (t_1, t_2)$, the conjugate point.

For the *critical damping case* $\alpha = 2\sqrt{\beta}$ the general solution such that $h(t_1) = 0$ is

$$h(t) = Ce^{-\sqrt{\beta}t}(t - t_1). \tag{16}$$

There is no non-trivial solution $h$ that vanishes also at $t \in (t_1, t_2)$ — *no conjugate points*. The same holds for the *overdamping case* $\alpha > 2\sqrt{\beta}$, where the solution that vanishes at $t_1$ is

$$h(t) = Ce^{-\frac{\alpha t}{2}}\left(e^{\frac{1}{2}\sqrt{\alpha^2-4\beta}t} - e^{\frac{1}{2}\sqrt{\alpha^2-4\beta}(2t_1-t)}\right). \tag{17}$$

For the *underdamping case* $\alpha < 2\sqrt{\beta}$, the picture gets more similar to the vanishing damping case (Sec.3.1). The solution under $h(t_1) = 0$ is

$$h(t) = Ce^{-\frac{\alpha t}{2}}\left(\sin\left(\frac{\sqrt{4\beta - \alpha^2}}{2}t\right) - \tan\left(\frac{\sqrt{4\beta - \alpha^2}}{2}t_1\right)\cos\left(\frac{\sqrt{4\beta - \alpha^2}}{2}t\right)\right). \tag{18}$$

Hence all points $t > t_1$ conjugate to $t_1$ satisfy $t = t_1 + 2k\pi/\sqrt{4\beta - \alpha^2}$ for $k \in \mathbb{N}$. Therefore for any $t_2 > t_1 + 2\pi/\sqrt{4\beta - \alpha^2}$ there exists a conjugate point $t \in (t_1, t_2)$.

**Theorem 9** (Global optimality of overdamped Nesterov, suboptimality of accelerated Nesterov). *Let $f : \mathbb{R}^d \to \mathbb{R}$ be a strongly convex quadratic, and let $X : \mathbb{R} \to \mathbb{R}^d$ be a solution to the ODE $\ddot{X} + \alpha\dot{X} + \nabla f(X) = 0$. For $0 \leq t_1 < t_2$, consider the action $J[Y] = \int_{t_1}^{t_2} L_\alpha(Y, \dot{Y}, t)dt$. If $\alpha \geq 2\sqrt{\beta}$, where $\beta$ is the max. eigenvalue of the Hessian of $f$, then Nesterov's path minimizes $J$ over all curves s.t. $Y(t_1) = X(t_1)$ and $Y(t_2) = X(t_2)$. Else (e.g. acceleration setting $\alpha \approx 2\sqrt{\mu}$), Nesterov's path is optimal only for $|t_2 - t_1| \leq 2\pi/\sqrt{4\beta - \alpha^2}$ and otherwise is a saddle point.*

*Proof.* As for the proof of Thm. 5, the condition on conjugate points has to hold for each eigendirection separately. We conclude by noting that eigenvalues are in the range $[\mu, \beta]$. $\square$

For the underdamping case, we give a concrete example for $\alpha = \beta = 1$, to show the saddle point nature. Consider the finite-norm perturbation $h(t) = \sin(k\pi(t - t_1)/(t_1 - t_2))$, where $k \in \mathbb{N}$. Then,

$$\delta^2 J[\gamma](\sigma h) = \sigma^2 e^{2t_1}\left(\frac{k^2 e^{-(t_2-t_1)}\pi^2(e^{t_2-t_1} - 1)(2k^2\pi^2 - (t_2 - t_1)^2)}{(t_2 - t_1)^2(4k^2\pi^2 + (t_2 - t_1)^2)}\right). \tag{19}$$

Hence, for any $t_2 - t_1 > \sqrt{2}k\pi$, it holds that $\delta^2 J[\gamma](\sigma h) < 0$.

**Extending the optimality claims to $t_2 = \infty$ with $\Gamma$-convergence.** From an optimization viewpoint, the most interesting setting is to study the *action over the complete trajectory*, i.e. to consider $Y \in \mathcal{C}^1([t_1, \infty), \mathbb{R}^d)$ such that $Y(t_1) = X(t_1)$ and $Y(\infty) = x^*$, a minimizer. Prop. 7 and Thm. 9 show that the question of optimality in this case deserves a discussion *only in the extremely overdamped case* $\alpha \geq 2\sqrt{\beta}$, where minimality is guaranteed for any time interval. A careful study of the infinite-time setting would require the theory of $\Gamma$-convergence (Braides et al., 2002). The usual pipeline consists in defining a sequence of problems $J_k$, on intervals $[t_1, t_2^k]$, with $t_2^k \to \infty$ as $k \to \infty$. Under the assumption that each $J_k$ admits a global minimizer (only true for the overdamped case), one can study convergence of $J_k^* = \min\{J_k[Y] : Y \in \mathcal{C}^1([t_1, t_2^k], \mathbb{R}^d)\}$ to $J_\infty^* = \min\{J_\infty(Y) : Y \in \mathcal{C}^1([t_1, \infty), \mathbb{R}^d)\}$. While existence of $J_\infty^*$ and $J_\infty$ is not trivial in general, for our setting the pipeline directly yields minimality of overdamped Nesterov's path until $\infty$.

## 5 Discussion of the main findings and directions for future research

In this section, we summarize the results of Sec. 3 & 4 and discuss some implications that our analysis delivers on the geometry of accelerated flows in the convex and strongly convex setting.

We summarize below the main high-level findings of our theoretical analysis:

1. The *optimality* of Nesterov's path for minimization of the action corresponding to the Bregman Lagrangian is strictly linked to the *curvature* around the minimizer reached by the flow.

2. As the maximal curvature $\beta$ increases, it gets increasingly difficult for *accelerated* flows to minimize the action over long integration intervals: both the accelerated ODEs $\ddot{X} + 3/t\dot{X} + \nabla f(X) = 0$ and $\ddot{X} + 2\sqrt{\mu}\dot{X} + \nabla f(X) = 0$ are optimal only for intervals of length $\propto 1/\sqrt{\beta}$.

3. If Nesterov's path does not minimize the action, there does not exist a "better" (through the eyes of the action) algorithm, as the functional gets unbounded from below (Sec. 3.3).

4. This suboptimality is *due precisely to the oscillations* in the accelerated paths. In contrast, as long as each coordinate in parameter space decreases monotonically (as it is the case for gradient descent), Nesterov's path is optimal. See also O'donoghue and Candes (2015).

5. Hence, Nesterov's method with very high damping $\alpha > 2\sqrt{\beta}$ — which does not oscillate and hence does not lead to acceleration (see Fig. 3) — minimizes the action.

In a nutshell, locally Nesterov's method does indeed optimize a functional over curves. However, this property breaks down precisely when the geometry gets interesting — i.e. when the loss evolution is non-monotonic. Since acceleration is a global phenomenon, i.e. is the cumulative result of many consecutive oscillations (see Fig. 3), our results suggest that the essence of *acceleration cannot be possibly captured by the minimization of the action relative to the Bregman Lagrangian*.

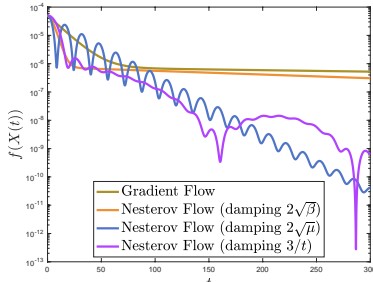

**Non-uniqueness of the Lagrangian.** A possible reason for the non-optimality of Nesterov's path is, simply put — that we are not looking at the right action functional. Indeed, there are many Lagrangians that can generate Nesterov ODE. Let $F(Y,t)$ be any function which does not involve the velocity, then it is easy to see that the Lagrangian $L$ is equivalent to

$$\tilde{L}(Y, \dot{Y}, t) = L(Y, \dot{Y}, t) + \left\langle \dot{Y}, \frac{\partial F}{\partial Y}(X, t) \right\rangle + \frac{\partial F}{\partial t}(X, t). \tag{20}$$

Figure 3: Optimization of potential $f(x, y) = 0.02x^2 + 0.0004y^2$, where $2e - 2 = \beta \gg \mu = 3e - 4$. Non-monotonic trajectories (i.e. the accelerated curves) minimize the action only for short time intervals. Simulation with Runge-Kutta 4 integration.

This simple fact opens up new possibilities for analyzing and interpreting Nesterov's method using different functionals — which perhaps have *both a more intuitive form and better properties*.

**Higher order ODEs.** On a similar note, it could be possible to *convexify the action functional* by considering a logical OR between symmetric ODEs, e.g $(\frac{d^2}{dt^2} + \alpha\frac{d}{dt} + \beta)(\frac{d^2}{dt^2} - \alpha\frac{d}{dt} + \beta)X = 0$. Such tricks are often used in the literature on dissipative systems (Szegleti and Márkus, 2020).

**Noether Theorem.** In physics, the variational framework is actually never used to claim the minimality of the solution to the equations of motion. Its power relies almost completely in the celebrated Noether's Theorem (Noether, 1918), which laid the foundations for modern quantum field theory by linking the symmetries in the Lagrangian (and of the Hamiltonian) to the invariances in the dynamics. Crucially, for the application of Noether's Theorem, one only needs the ODE to yield a stationary point for the action (also saddle points work). Coincidentally, while finalizing this manuscript, two preprints (Tanaka and Kunin, 2021; Głuch and Urbanke, 2021) came out on some implications of Noether's Theorem for optimization. However, we note that these works do not discuss the direct link between Noether's Theorem and acceleration, but instead study the interaction between the symmetries in neural network landscapes and optimizers. While some preliminary implications of Noether's theorem for time-rescaling of accelerated flows are discussed in (Wibisono et al., 2016), we suspect that a more in-depth study could lead, in combination with recent work on the Hamiltonian formalism (Diakonikolas and Jordan, 2019), to substantial insights on the hidden invariances of accelerated paths. We note that finding these invariances might not be an easy task, and requires a dedicated work: indeed, even for simple linear damped harmonic oscillators (constant damping), invariance in the dynamics can be quite complex (Choudhuri et al., 2008).

# 6   Conclusion

We provided an in-depth theoretical analysis of Nesterov's method from the perspective of calculus of variations, and showed that accelerated paths only minimize the action of the Bregman Lagrangian locally. This suggests that further research is needed for understanding the underlying principle of generation for acceleration — which has been an open question in optimization for almost 40 years (Nesterov, 1983). To this end, we proposed a few concrete directions for future work.

## Acknowledgments and Disclosure of Funding

The authors thank Prof. Luca Martinazzi for his helpful discussions. This work was partially funded in part by the French government under management of Agence Nationale de la Recherche as part of the "Investissements d'avenir" program, reference ANR-19-P3IA-0001(PRAIRIE 3IA Institute). Hadi Daneshmand acknowledges support from the European Research Council (grant SEQUOIA 724063).

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
