# OpenReview forum: "Rethinking the Variational Interpretation of Accelerated Optimization Methods"
_NeurIPS.cc/2021/Conference — NeurIPS 2021 Poster_

### Official Review · Reviewer_buiE · 2021-07-15

**Rating:** 7
**Confidence:** 2

**Summary:**

This paper considers Nesterov’s accelerated method from a novel perspective and sheds lights onto its variational interpretation. This paper reads really well and opens further avenues for understanding the acceleration phenomenon.

**Limitations And Societal Impact:**

No societal impact.

**Main Review:**

The paper starts from an observation that optimality of the Nesterov’s method has not been rigorously demonstrated in previous works. This is an important issue, as if the Nesterov’s method (and other accelerated methods) does not minimise the proposed action functionals, then their interpretation becomes unclear. This paper does a good job explaining this and even proposing some alternative action functionals that would give rise to the Nesterov’s method.

While I haven’t checked the proofs (as they go beyond my knowledge of this area), I found the paper as an insightful read and an important contribution to the area.

1) Can authors expand on the comments that the action functional is not unique - therefore, it is possible to get more intuitive action functionals for improving the understanding of accelerated methods? Is there an action functional where the Nesterov arises as the optimal?

2) Also can authors clarify what is the use and advantage of convexifying the action functional? What kind of properties and advantages does it bring in terms of interpreting acceleration or developing new accelerated methods?

3) I believe that the small paragraph written about Noether Theorem wasn’t clear. Actually, it wasn’t possible to understand how it relates to the rest of the work and why it is written. It would be good to clarify.

**Time Spent Reviewing:**

6

---

> ### Author Response · Authors · 2021-08-09
> **Answer to your questions**
>
> We thank the reviewer for the nice comments on our paper – we are very happy to share our findings and techniques with the NeurIPS community!
>
> We reply below to your interesting questions:
>
> 1) This question is very challenging and is totally in line with our direction of future work. We think it is very difficult to find a Lagrangian that is both interpretable and leads to minimality of Nesterov’s path. The most concrete idea we have is to consider a differential equation of a higher degree, as explained in line 297. Using this equation, however, implies that Nesterov’s algorithm has to be replaced with something “more complex” (fourth-order ODE) – yet hopefully still accelerated. This direction of research is very intriguing and hence deserves to be explored in a separate publication.
>
> 2) Convexification of the functional would guarantee minimality: If Nesterov’s path is an extremum for a convex action, then it also minimizes the action.
>
> 3) Thank you for your feedback on this paragraph – we will make our best efforts to make it clearer. Here is the motivation: in the last months we talked to a lot of physicists, asking for opinions on the role of the Lagrangian in the laws of physics. Their response was “Noether Theorem”. Indeed, in modern quantum field theory, this theorem provides compelling motivation for the assumptions in modern field theory (see e.g. Landau Lifshitz's first book). In essence, N’s theorem allows mapping structural symmetries of a physical system to conserved quantities in the dynamics. This could be in principle very important in optimization since conserved quantities are linked to Lyapunov functions – often used to derive convergence rates. As a result, Noether’s theorem would ideally allow deriving the rate of Nesterov’s method from the symmetries in the Lagrangian. However, we have to note that, carrying out these computations does not necessarily lead to nice results:  (Choudhuri et al., 2008) explored the symmetries of damped harmonic oscillators (i.e. one-dimensional Nesterov's paths) and found the corresponding conserved quantities using Noether’s theorem – these are very complex and not easy to interpret. We plan to explore the topic this autumn, it is not easy but hopefully gives nice results, fingers crossed!

---

### Official Review · Reviewer_4GWq · 2021-07-19

**Rating:** 4
**Confidence:** 4

**Summary:**

In prior work, Su et al. (2016) introduced a continuous-time modelling of Nesterov's accelerated gradient descent (AGD) and analyzed its property. Wibisono et al. (2016) introduced the Bregman Lagrangian, which obtains Su et al.'s acceleration ODE via the principle of "least action" and establishes the connection of AGD with classical Lagrangian and Hamiltonian mechanics of mathematics physics. In this work, the authors point out an (in my view) significant misconception that the acceleration ODE arises from a "least action" path; rather, it is merely a stationary point.

**Main Review:**

The authors provide the first rigorous study of the action relative to Bregman Lagrangian and characterize (the non) minimality of Neterov's path. The clarity of this paper is good. I enjoyed reading the paper and learned a good deal from the discussion of prior work. The misconception that the authors point out is one that I also held myself, so I am glad to be corrected. I am also under the impression that the original authors of the Bregman Lagrangian work shared this misconception, so I imagine this work will serve to prevent the misconception from being propagated by them as well.

However, after reading the paper, I am left with the feeling of "so what"? This work provides a valuable correction on how we interpret the prior analyses, but it doesn't invalidate any of the prior concrete results nor immediately inform us of what to do differently. The authors mention that the power of the variational framework lies almost completely in Noether Theorem. Perhaps fleshing this out would lead to, I suspect, a really interesting line of work. However, the paper, as is, I feel does not contain enough interesting content to be published at NeurIPS.


**Time Spent Reviewing:**

4

---

> ### Author Response · Authors · 2021-08-09
> **Remarks on our contribution**
>
> We thank the reviewer for the very positive comments, we are glad you found our paper significant and that you learned a lot from reading it!
>
> - *“so what?”*
>
> Our paper points out and builds upon an **important misconception on possibly the main selling point of – the second most cited – paper on the ODE interpretation of Nesterov’s method**. To do this, we carefully studied the variational problem: it took months and several discussions with top mathematical analysis professors. We also remark that our results on the (non) optimality of Nesterov’s path (e.g. Thm 9) are quantitative (bound on the damping is derived), novel, and can be potentially used for further analysis.
>
> However, you are right: our main takeaway is a negative result: Nesterov sadly does not minimize the action globally. **Given the negative nature of our result, we understand and totally relate with your feeling “so what?”, but we like to kindly point out that, fortunately for the development of the field, negative results are also published at top conferences**.
>
> We therefore absolutely do not think our contribution is marginal: we (the authors) have years of experience in working on the ODE interpretation of Nesterov’s method, and while talking to colleagues at many conferences (including Neurips, ICML, AISTATS) found that many researchers believe the variational interpretation proves Nesterov’s path is optimal in function space (see also talks linked in our reply to Reviewer 7bjp). This is also hinted at in many recent publications (see line 71), and we prove this to be wrong.
>
> - *Noether Theorem*
>
> We concluded the paper with many ideas for future research – to end on a positive note – but felt that the exploration of these topics deserves (potentially) a separate publication. Indeed, it does take 8 dense pages to fully explore the variational problem associated with Nesterov’s method, and adding an additional topic would hurt the quality of a 9-pages writeup (which as you mention you enjoyed reading, thank you again for this comment). This is especially because, given the preliminary results on the application of Noether Theorem in damped harmonic oscillators (Choudhuri et al., 2008), we expect the discussion to be complex, hence adding this to the paper would hurt the focus of our work on the minimality of Nesterov’s path.
>
> - *Remark on reviewer's flexibility*
>
> We hope that the reviewer can find some additional flexibility in the final evaluation of our (maybe somewhat unusual) piece of work: as you also mention this would “prevent the significant misconception from being propagated”. We think **this misconception in the literature should be pointed out as soon as possible**. We think this, combined with your excitement in reading the paper, justifies the visibility the NeurIPS publications offer: **setting the record straight is our contribution**. We will make this clearer for a potential camera-ready version.

---

### Official Review · Reviewer_7bjp · 2021-07-26

**Rating:** 6
**Confidence:** 4

**Summary:**

The Lagrangian variational characterization of continuous time trajectories that converge to the global minimum of a convex objective function at a rate faster that $O(1/t)$ is an area of active research, and this paper addresses the question of whether the Euler-Lagrange trajectories induced by the Bregman Lagrangian are minimizers of the action integral. This question is motivated by a statement made in Michael Jordan's 2018 ICM proceedings paper, which alludes to the trajectories optimizing (as opposed to extremizing) the action integral.

The main contributions of the paper are as follows:

1. Proving that the action integral is a minimum only for sufficiently small time intervals (Theorem 5), which appears to be a consequence of the well known result in the calculus of variations that local coercivity induces local minimum.
2. Constructively proving that the action for the Nesterov trajectory is suboptimal in the quadratic setting (Proposition 7).
3. A proof that the overdamped Nesterov trajectory minimizes the action integral for arbitrarily long time intervals, whereas the accelerated Nesterov trajectory is a saddle (Theorem 9).


**Main Review:**

The Lagrangian variational characterization of continuous time trajectories that converge to the global minimum of a convex objective function at a rate faster that $O(1/t)$ is an area of active research, and this paper addresses the question of whether the Euler-Lagrange trajectories induced by the Bregman Lagrangian are minimizers of the action integral. This question is motivated by a statement made in Michael Jordan's 2018 ICM proceedings paper, which alludes to the trajectories optimizing (as opposed to extremizing) the action integral.

The main contributions of the paper are as follows:

1. Proving that the action integral is a minimum only for sufficiently small time intervals (Theorem 5), which appears to be a consequence of the well known result in the calculus of variations that local coercivity induces local minimum.
2. Constructively proving that the action for the Nesterov trajectory is suboptimal in the quadratic setting (Proposition 7).
3. A proof that the overdamped Nesterov trajectory minimizes the action integral for arbitrarily long time intervals, whereas the accelerated Nesterov trajectory is a saddle (Theorem 9).

The proofs appear to be sound, but nothing in the Lagrangian variational approach to mechanics necessitates that the action integral is a minimizer (as opposed to extremizer) over arbitrarily long time intervals, so the fact that the Nesterov accelerated trajectory is generally a saddle point as opposed to minimizer of the action integral is hardly surprising or noteworthy. In honesty, I think the paper under consideration is fixating on a misstatement that has never served as the theoretical justification for the variational approach to accelerated optimization, which is why this issue has never really been studied rigorously.

There is nothing technically wrong with the results in the paper, but the underlying motivation for the paper appears to be somewhat misguided. In particular, for the paper to be truly significant, I would like the authors to point out instances in which the variational approach to accelerated optimization relies critically on the incorrect assumption that the action integral is minimized as opposed to being extremized.

Some additional comments: Since Theorems 1-4 are well known in the calculus of variations, and are presented without proof, the authors should cite a reference for these results.

**Time Spent Reviewing:**

2

---

> ### Author Response · Authors · 2021-08-09
> **Our main contribution is setting the record straight on the (arguably misguided) interpretation of the Variational Framework**
>
> We thank the reviewer for the positive comments about our work and for the interesting remark on the practical use of the (imprecise) assumption of minimality.
>
> - *“point out instances in which the variational approach relies critically on the incorrect assumption”*
>
> We thank the reviewer very much for this request, which deserves to be answered thoroughly, and that we are happy to further discuss at any point in the comments.
>
> The main “application” of the variational formulation is arguably the intuition it provides: namely, it shows how Nesterov’s path is linked to a meta-optimization problem in the space of curves. This construction (Wilson et al.) is extremely elegant, and could in principle provide an explanation for the mysterious nature of the acceleration phenomenon (many papers are devoted to this in recent years). To put it in other words, the minimality of Nesterov’s path is used as a rhetoric strategy (see evidence in videos linked below) to motivate the use of the ***variational framework as a fundamental step towards explaining the acceleration mechanism***.
>
> ***Our contribution is to show that this intuition is unfortunately wrong*** in many cases of interest – i.e. we provide a negative result.
> To sum it up: our contribution in this paper is setting the record straight on the link between the variational formulation and the acceleration phenomenon.
>
> ***Evidence for the (imprecise) claim of Nesterov’s meta-minimality*** over paths is most apparent in two popular talks by professor Jordan, here are the recordings: https://www.youtube.com/watch?v=VE2ITg_hGnI [Simons Institute], https://www.youtube.com/watch?v=wXNWVhE2Dl4 [ICM 2018]. In particular, I would point to minute 17:50 of the Simons Institute talk, where the minimality of Nesterov’s path is explicitly stated in the slides. If the reviewer has some time, I would encourage he/she to watch part of the videos: it is repeated many times that the main conceptual contribution of the variational framework is to unveil the nature of acceleration through an argument around minimality over paths.
>
> In our paper, we make the discussion precise, which we think is a very valuable contribution given the influence of these papers (Wibisono et al., 2016) has more than 300 citations, which are a lot in this line of research – second most cited paper), and given the popularity of Prof. Jordan talks (almost 20k views for the two linked above combined).
>
> - *“the authors should cite a reference for CoV results”*
>
> Please see lines 89-90.  We will recall these references also in the dedicated sections.

---

> > ### Comment · Reviewer_7bjp · 2021-08-14
> > **Thank you for your response**
> >
> > Thank you for your response, I have read it and will take it into consideration.

---

### Official Review · Reviewer_XNyg · 2021-07-28

**Rating:** 4
**Confidence:** 5

**Summary:**

The authors try to understand the low-resolution ODE for Nesterov accelerated gradient descent in convex case from the variational perspective. Meanwhile, they find the ODE is not the minimizer of the action functional, but the saddle. From mathematical perspective, it is a new and novel point to investigate.

**Limitations And Societal Impact:**

This paper needs to show more convince.

**Main Review:**

The topic in this paper is very interesting. Understanding the Nesterov' method from continuous perspective is a very attractive topic. But the content is so few that it is hard to persuade the readers the framework makes sense, such as how about other ODEs and high-resolution ODEs in (Understanding the Acceleration Phenomenon via High-Resolution Differential Equations). Moreover, the purpose of the authors to investigate the low-resolution ODE is not clear for readers, which looks to start from a pure mathematical view. If yes, please show more results about all related ODEs; if no, please show more motivation. The quality is not enough. Furthermore, the authors need to show significance of this paper. Finally, if the authors can show more convince, I would like to change my grade.


**Time Spent Reviewing:**

25

---

> ### Author Response · Authors · 2021-08-09
> **Results for related ODEs (such as high-resolution) directly follow (proof below)**
>
> We thank the reviewer for finding our results interesting. We answer your comments below
>
> -  *“need to show results for more ODEs”*
>
> Thanks a lot for this very interesting suggestion to improve our work! ***It is very easy to show that a similar negative result (i.e. Nesterov’s ODE is not optimal in the space of paths) holds for the Lagrangian associated with the high-resolution Nesterov’s method***. We chose to present the results for the low-resolution equations simply because these are the objects studied in Wibisono et al. (2016).
> The reasoning behind the proof for the high-res odes is conceptually equivalent – we will add a note and a dedicated section in the appendix. ***Here is a compressed reasoning to convince you***: consider the high-res ODE for convex acceleration, with damping of the form $3/t + \sqrt{s} \nabla^2 f(X(t))$. As time goes to infinity, this damping converges to a small positive quantity, leading to oscillations in the path. In Theorem 9 and line 279 we prove that these oscillations are precisely the reason for the suboptimality of the ODE path – one needs very high damping for optimality. The same reasoning (even simpler) holds for the high-resolution strongly-convex ode with damping $\mu +  \sqrt{s} \nabla^2 f(X(t))$.
>
>
> - *“the content is so few that it is hard to persuade the readers the framework makes sense”*
>
> We first like to kindly point out that we do not invent this framework, but the variational perspective was introduced by Wibisono et al. (2016) – this is the second most cited paper in the continuous-time acceleration literature.
> Secondly, we think that the results we present are highly surprising, mathematically involved, and challenge the intuition behind many papers in the acceleration literature (see line 71). In these papers, the variational framework is widely utilized as a major principle to design powerful accelerated optimizers under different settings and is therefore of huge theoretical interest.
>
> It took us months to fully understand the necessary theorems on calculus of variations and to successfully apply them to this setting. We also talked to many top professors in mathematical analysis to make sure our claims are sound and formal. We agree that the results do not need pages and pages of calculations, but are conceptually very involved and rely on deep results in calculus of variations. Reviewers 4GWq and buiE both agree that the results are important in this line of research.
>
>
>
>
> -  *“show more significance”*
>
> Our paper is focusing on the derivation of one crucial result, which as Reviewers 4GWq and buiE mention “is an important contribution” and “points out a significant misconception”. We do elaborate on the importance of this result in the introduction but will make the discussion stronger, as you suggest, for a potential camera-ready version.
>
> - *To sum this up..*
>
> We hope that the reviewer would positively reconsider our submission, since (1) we showed how our results can be extended to the high-resolution and even more general cases in just a few lines, and (2) we are willing to modify the introduction to match the excitement level shown by all reviewers.

---

> ### Author Response · Authors · 2021-08-23
> **Update**
>
> Dear reviewer,
>
> Since you mentioned that you will reconsider your score if we address your main concerns, we would be grateful to get some feedback on the answers we provided. We are happy to clarify any point if needed.
>
> Thank you very much, The authors

---

### Decision · Program_Chairs · 2021-09-27

**Decision:**

Accept (Poster)

**Comment:**

This paper discusses a misconception (or implicit assumption) from some popular recent work that links Nesterov acceleration to continuous time equations.  There was a very robust discussion between the reviewers and author(s), and also privately among the reviewers.  We also solicited an extra review, and the AC and SAC had involved discussions.

Generally, the reviewers liked the paper, and there were no fatal flaws or mistakes.  The writing is nice, and it's certainly not a hostile paper.  The main argument against accepting the paper is that it is not clear the misconception has led to incorrect research, and it's also not clear how influential the original arguments have been.

However, the paper offers more than just pointing out the misconception (it discusses regimes where Nesterov's method is optimal) and makes some nice connections.  It's an enjoyable read, and useful to anyone interested in modern optimization. Furthermore, the original misconception had been repeated in high-profile venues (like a plenary talk at the ICM).  While it's not certain that this paper is currently needed in order to fix state-of-the-art research, this paper does contribute to the literature and may be useful in the future.

Overall, this is a nice paper, and has a chance of making an impact, hence we're pleased to suggest its acceptance.